# Differential Affinity Chromatography Coupled to Mass Spectrometry: A Suitable Tool to Identify Common Binding Proteins of a Broad-Range Antimicrobial Peptide Derived from Leucinostatin

**DOI:** 10.3390/biomedicines10112675

**Published:** 2022-10-23

**Authors:** Joachim Müller, Ghalia Boubaker, Dennis Imhof, Kai Hänggeli, Noé Haudenschild, Anne-Christine Uldry, Sophie Braga-Lagache, Manfred Heller, Luis-Miguel Ortega-Mora, Andrew Hemphill

**Affiliations:** 1Institute of Parasitology, Vetsuisse Faculty, University of Bern, Länggass-Strasse 122, 3012 Bern, Switzerland; 2Proteomics and Mass Spectrometry Core Facility, Department for BioMedical Research (DBMR), University of Bern, 3012 Bern, Switzerland; 3SALUVET, Animal Health Department, Faculty of Veterinary Sciences, Complutense University of Madrid, Ciudad Universitaria, 28040 Madrid, Spain

**Keywords:** animal experimentation, drug targets, mass spectrometry, modeling, peptides as drugs, side effects

## Abstract

Leucinostatins are antimicrobial peptides with a broad range of activities against infectious agents as well as mammalian cells. The leucinostatin-derivative peptide ZHAWOC_6027 (peptide 6027) was tested in vitro and in vivo for activity against the intracellular apicomplexan parasite *Toxoplasma gondii*. While highly efficacious in vitro (EC50 = 2 nM), subcutaneous application of peptide 6027 (3 mg/kg/day for 5 days) in mice experimentally infected with *T. gondii* oocysts exacerbated the infection, caused mild clinical signs and elevated cerebral parasite load. Peptide 6027 also impaired the proliferation and viability of mouse splenocytes, most notably LPS-stimulated B cells, in vitro. To identify common potential targets in *Toxoplasma* and murine splenocytes, we performed differential affinity chromatography (DAC) with cell-free extracts from *T. gondii* tachyzoites and mouse spleens using peptide 6027 or an ineffective analogue (peptide 21,358) coupled to N-hydroxy-succinimide sepharose, followed by mass spectrometry. Proteins specifically binding to peptide 6027 were identified in eluates from the peptide 6027 column but not in peptide 21,358 nor the mock column eluates. In *T. gondii* eluates, 269 proteins binding specifically to peptide 6027 were identified, while in eluates from mouse spleen extracts 645 proteins specifically binding to this peptide were detected. Both datasets contained proteins involved in mitochondrial energy metabolism and in protein processing and secretion. These results suggest that peptide 6027 interacts with common targets in eukaryotes involved in essential pathways. Since this methodology can be applied to various compounds as well as target cell lines or organs, DAC combined with mass spectrometry and proteomic analysis should be considered a smart and 3R-relevant way to identify drug targets in pathogens and hosts, thereby eliminating compounds with potential side effects before performing tedious and costly safety and efficacy assessments in animals or humans.

## 1. Introduction

Initial steps in drug development against infectious diseases are the primary assessment of compound efficacy and cytotoxicity against mammalian cells. Importantly, the development of in vitro culture models has enabled researchers to identify a plethora of novel drug candidates that exhibit a favorable therapeutic window, meaning low half-maximal effective concentrations (EC_50_s) against infectious agents and low or no cytotoxicity against host cells. In the typical drug development workflow, these initial in vitro assessments are followed up by experimental studies in rodents prior to moving forward to larger animals and subsequently humans. Typically, studies on drug absorption, distribution, metabolism and excretion, but also pharmacokinetic properties (PK), stability and efficacy, as well as toxicity, are performed in pregnant and non-pregnant rodent models. Our own research has largely focused on anti-infective agents to be used against *Neospora caninum* [1] and *Toxoplasma gondii* [2], two closely related protozoan parasites (phylum Apicomplexa [3]) causing abortion in humans and farm animals. Unfortunately, in too many cases, drugs with promising results in cell-culture-based assays were either ineffective in the mouse model or even caused adverse side effects, such as fetal malformations leading to abortion or high neonatal mortality in pups [4,5,6]. This occurred irrespective of whether the compounds were issued from a classical screening (e.g., drug repurposing) or from target-based development [7].

To be compliant to the 3R principles of animal experimentation (reduce–replace–refine; [8]), this frequent failure in translating promising results from in vitro models to the in vivo situation demands a modification of the prevailing experimental set-up. After the determination of the effects on parasite proliferation and host cell toxicity, but prior to testing a given set of compounds in a suitable in vivo model, the potential impact on the proliferative capacity and viability of relevant host cells, in particular immune cells such as in vitro cultured and stimulated splenocytes, should be investigated [9]. Since *Neospora* and *Toxoplasma* cause disease during pregnancy, the effects of interesting candidate compounds on zebrafish embryo development can be assessed [6]. This approach includes, however, animal experimentation and has only limited predictive value concerning effects on mammalian embryos [6]. A less ambiguous and tedious way of eliminating compounds with severe side effects may reside in the identification of potential targets in essential pathways in host cells. In earlier studies, we have successfully identified common targets of thiazolides, a class of broad anti-infective and anti-cancer drugs, in protozoan parasites [10,11], as well as host cells [12,13], by affinity chromatography, followed by gel electrophoresis and the identification of major bands by mass spectrometry (MS). In recent studies [14,15], we have alleviated the limitations of this method by subjecting cell-free extracts from parasites and suitable host cells to differential affinity chromatography (DAC) coupled to MS. This approach allows the identification of entire proteomes binding to compounds of interest but not to (closely related) ineffective compounds and not to single proteins anymore [14]. Thereby, common pathways in parasites, as well as in mammalian hosts, which are touched by a given (class of) compounds, can be identified [15]. In an ideal situation, compounds interfering with essential pathways such as protein biosynthesis will be discarded or improved before subjecting them to in vivo trials.

Peptides are highly flexible with respect to their structure, and their synthesis is easy. Therefore, they constitute an ideal class of compounds for antimicrobial drug development. Recently, using *Trypanosoma brucei* (Alveolata) as a model organism, derivatives of the antibiotic peptide leucinostatin [16], have been identified as powerful antiprotozoal agents most likely affecting the stability of the inner mitochondrial membrane [17]. In the present study, we present the effects of one of the most promising compounds of this study, the peptide ZHAWOC_6027 (henceforth referred to as peptide 6027), on *Toxoplasma gondii* and mouse splenocytes. While peptide 6027 was highly active against *T. gondii* tachyzoites in vitro at concentrations that did not affect human foreskin fibroblast host cells, in vivo studies in mice orally receiving peptide 6027 showed that the compound was ineffective and that treatments were prone to inducing adverse side effects. Subsequently, it was found that peptide 6027 also affected the proliferation and viability of in vitro cultured murine splenocytes. Using DAC with peptide 6027 and a related but ineffective peptide, we identified proteins specifically interacting with peptide 6027 involved in essential cellular and metabolic pathways in both parasite and host splenocytes, thereby explaining the failure of this peptide in vivo.

## 2. Materials and Methods

### 2.1. Tissue Culture Media, Biochemicals and Compounds

Cell culture media was purchased from Gibco-BRL (Zürich, Switzerland), and biochemical agents were procured from Sigma (St. Louis, MO, USA). The antimicrobial leucinostatin derivatives were synthesized at the Zürcher Hochschule für Angewandte Wissenschaft as described earlier [17].

### 2.2. In Vitro Culture, Collection of Parasite and Efficacy Assessments In Vitro

*Toxoplasma gondii* RH wildtype tachyzoites and RH tachyzoites expressing *E. coli* beta-galactosidase were grown in Vero cells as previously described [13,18]. For affinity chromatography, pellets of RH wildtype tachyzoites were stored at −80 °C until processing. For drug efficacy tests, RH beta-galactosidase tachyzoites were used directly after harvest. Inhibition tests on *T. gondii* were done using human foreskin fibroblasts (HFF) as host cells as previously described [14]. The impact of antimicrobial peptide 6027 on HFF vitality was determined by resazurin reduction assay [6]. Inhibitory concentrations corresponding to 50% of the control values (IC_50_) were calculated as described [7].

### 2.3. In Vivo Assessment of Peptide 6027 for the Treatment of Mice Experimentally Infected with T. gondii Oocysts

Animal experiments were approved by the Animal Welfare Committee of the Canton of Bern (license BE117/2020), and mice were handled in strict accordance with practices to minimize suffering. Twenty-four female CD1 mice, 6 weeks of age, were purchased from Charles River (Sulzberg, Germany) and were maintained in a common room under controlled temperature and a 14 h/10 h light/dark cycle in the animal facility for two weeks for adaptation prior to the experiments. They were randomly allocated to three groups, and 2 groups (16 animals) were orally infected with 200 *T. gondii* oocysts (TgShSp1; [19]). At 2 days post-infection (p.i.), the mice in the group designated 6027 were treated by the subcutaneous (s.c.) application of peptide 6027 formulated in 20%/50%/30% of DMSO/PEG/PBS, respectively, at 3 mg/kg/day for 5 days; group C+ received only DMSO/PEG/PBS without a compound, and group C− was neither infected nor treated with peptide 6027. During and following treatments, animals were closely monitored for clinical signs. On day 30 p.i., all animals were euthanized and dissected.

Blood and brain samples, as well as spleens, were collected. Total IgG was measured as described [19]. The quantification of the cerebral parasite load was done by real-time PCR as described earlier [20,21]. For DNA isolation from brain, a NucleoSpin DNA RapidLyse Kit (Macherey-Nagel, Oensingen, Switzerland) was used according to standard protocols, and DNA concentrations were quantified using the QuantiFluor double-stranded DNA (dsDNA) system (Promega, Madison, WI, USA). Quantitative real-time PCR was performed as previously described [21]. Spleens from all three groups were processed and splenocytes isolated as described [22]. Cultured splenocytes were re-stimulated with concanavalin A (5 µg/mL; Sigma, St. Louis, MO, USA), with soluble *T. gondii* protein extract (20 µg/mL) or remained unstimulated as a negative control. After 72 h of stimulation, supernatants were collected and stored at −80 °C. Levels of IFN-γ in culture supernatants were determined by ELISA (BD OptEIATM Mouse ELISA Set, LifeSpan, Biosciences Inc., Seattle, WA, USA) as previously described [22]. Parasite burdens, antibody titers and IFN-γ were compared between groups by the non-parametric Kruskal–Wallis test, followed by a Mann–Whitney-U test. Statistical analysis was performed using Graphpad Prism version 9.3.1 for MacOS (GraphPad Software, La Jolla, CA, USA, www.graphpad.com, accessed on 3 June 2021).

### 2.4. Assessment of Susceptibility of Murine Splenocytes to Peptide 6027

Murine splenocytes were isolated from female BALB/c mice as previously described [9]. The viability of isolated cells was determined using a Trypan Blue dye exclusion test, and only preparations with >99% viable cells were used. The spleen cell preparation was suspended in RPMI 1640 medium, including 10% FCS, 0.05 mM 2-mercaptoethanol, 2 mM L-glutamine and 100 U of penicillin plus 50 mg of streptomycin per mL, and cells were distributed in polystyrene 96 well flat-bottom sterile plastic plates (Greiner Bio-One, Kremsmünster, Austria; HuberLab, Aesch, Switzerland) at 2 × 10^5^ cells/100 μL/well. For proliferation/viability assays, splenocytes were either left unstimulated or were stimulated with Concanavalin A (ConA, 5 μg/mL), lipopolysaccharide (LPS, 10 μg/mL), ConA plus peptide 6027 (0.1–2 µM) or LPS plus peptide 6027. Experiments were done in quadruplicate in 200 µL wells, and cultures were maintained at 37 °C/5% CO_2_ for 72 h. The proliferative responses of splenocytes were measured using a BrdU cell proliferation kit (QIA58, Merck Millipore, Burlington, MA, USA) according to the instructions provided by the manufacturer, and absorbance measurements were done at 450/540 nm in an EnSpire multilabel reader (Perkin Elmer, Waltham, MA, USA). To measure the impact on viability, resazurin (0.1 mg/mL) was added, and the fluorescence intensity was measured at 530/590 nm at 0, 1, 2, 3, 4 or 5 h. Differences were calculated by subtracting time point 0 values from each time point. Data are presented as mean of emission +/− SD for the indicated numbers. Data comparisons between groups were examined using a Student’s *t*-test.

### 2.5. Transmission Electron Microscopy (TEM)

HFF monolayers grown in T25 tissue culture flasks were infected with 1 × 10^7^ *T. gondii* RH tachyzoites in culture medium (4 h, 37 °C/5% CO_2_). Starting from 4 h post-infection, cultures underwent continuous treatment with 200 nM peptide 6027 or were left untreated. At 6, 24 and 48 h after the initiation of treatment, infected monolayers were fixed overnight at 4 °C in 100 mM sodium cacodylate buffer (pH 7.3) containing 2.5% glutaraldehyde. Subsequently specimens were washed in buffer and were post-fixed in osmium tetroxide, pre-stained, dehydrated and embedded in Epon812 epoxy resin as described [21]. Ultrathin sections were cut on a Reichert & Jung (Vienna, Austria) microtome and placed onto 300 mesh formvar–carbon coated grids (Plano GmbH, Wetzlar, Germany). Specimens were observed on a Philips CM12 TEM operating at 80 kV.

### 2.6. Protein Extraction and Differential Affinity Chromatography (DAC)

For protein extraction, frozen pellets of *T. gondii* RH tachyzoites or mouse spleens were resuspended in ice-cold extraction buffer, i.e., PBS containing 1% Triton X-100 and 1% of Halt proteinase inhibitor cocktail (ThermoFisher). Suspensions were vortexed thoroughly and centrifuged (13,000× *g* rpm, 10 min, 4 °C). Extractions were repeated twice. Three mL of extraction buffer was used in total. Supernatants were combined (approximately 2 mg of total protein) and subjected to affinity chromatography.

Matrices conjugated to peptide 6027 or to the ineffective analogue peptide 21,358 were produced by coupling 7 mg of each peptide to 1 mL of N-hydroxy-succinimide (NHS)-activated Sepharose 4 Fast flow (Cytiva, Marlborough, MA, USA) according to the manufacturer’s instructions. The column medium was then transferred to a chromatography column (Novagen, Merck, Darmstadt, Germany) and extensively washed with PBS-DMSO (1:1) and PBS in order to remove unbound compounds. The columns were stored in PBS containing 0.02% NaN_3_ at 4 °C.

Prior to affinity chromatography, mock columns prepared in the same way, but without peptide, were combined with either 6027 or 21,358 columns in tandem (mock first, then compound) and washed with 50 mL PBS equilibrated at 20 °C. Crude extracts (3 mL) prepared as described above were loaded with a flow rate of 0.25 mL/min. The column was washed with PBS until the baseline was flat (10 column volumes, corresponding to ca. 25 mL). Then, the columns were separated, and the binding proteins were eluted with 50 mM acetic acid (5 mL per column). The eluates were lyophilized and stored at −80 °C [14,15].

### 2.7. Proteomic Analysis of the Eluted Proteins by Mass Spectrometry

The lyophilized eluates were dissolved in 10 μL of 8 M urea and 0.1 M of Tris-HCl (pH 8); then, 1 μL of 0.1 M of Tris-HCl (pH 8) buffer containing 0.1 M of dithiothreitol was added, followed by incubation for 30 min at 37 °C and constant mixing at 600 rpm. This step was repeated with 1 μL of 0.5 M of iodoacetamide. Iodoacetamide was quenched by the addition of 5 μL 0.1 M of Tris-HC (pH 8) and the urea concentration further diluted to 4 M by the addition of 2 mM calcium dichloride in 20 mM Tris buffer. Proteins were digested for 2 h at 37 °C, followed by the addition of 1 μL of 0.1 μg/μL LysC sequencing grade protease (Promega), followed by the further dilution of urea to 1.6 M with the above calcium dichloride buffer and 1 μL of 0.1 μg/μL trypsin of sequencing grade (Promega). Digestion was completed by incubation overnight at ambient room temperature. Digestion was stopped with 2.5 μL of 20% (*v*/*v*) trifluoroacetic acid. After an incubation for 15 min at room temperature, the digest was spun for 1 min at 16,000 g, and the cleared supernatant was transferred to a HPLC vial for subsequent nano-liquid reversed phase chromatography coupled to tandem mass spectrometry, essentially as described earlier [23], with the exception that an Ultimate 3000 chromatograph coupled to a QExactive HF (Thermo Fisher Scientific, Bremen, Germany) was used. Each digest was analyzed three times by loading 5 μL and applying a 60 min gradient at a flow rate of 350 nl/min. The Full Scan method was set, with a resolution at 60,000, with an automatic gain control (AGC) target of 1E06 and a maximum ion injection time of 50 ms. The data-dependent method for precursor ion fragmentation was applied with the following settings: resolution 15,000, AGC of 1E05, maximum ion time of 110 milliseconds, mass window 1.6 *m*/*z*, collision energy 28, under fill ratio 1%, charge exclusion of unassigned and 1+ ions, and peptide match preferred, respectively.

The mass spectrometry data were processed with MaxQuant (v1.6.14.0) as described earlier [24]. Precursor and fragment mass tolerances were set to 10 ppm and 20 ppm, respectively, against a current protein sequence database release from toxodb.org for *Toxoplasma gondii* (ToxoDB-52_TgondiiRH88_AnnotatedProteins.fasta) or against Swissprot (www.uniprot.org) for *Mus musculus* (release version 2021_03). The MaxQuant search results were then further processed in R-studio. Alignments were performed using the Clustal Omega tool provided by the Expasy network (www.expasy.org). The protein–protein interactions of the affinoproteomes were analyzed using the STRING knowledge base and software tool (Swiss Institute of Bioinformatics, Lausanne, Switzerland; version 11.5; last accession 31 March 2022).

## 3. Results

### 3.1. The Antimicrobial Peptide 6027 Selectively Impairs the Proliferation and Structural Integrity of T. gondii Tachyzoites In Vitro

The antimicrobial peptide 6027 strongly affected the proliferation of *T. gondii* RH tachyzoites expressing *E. coli* beta-galactosidase as a reporter gene. The corresponding IC_50_ was 2.0 nM, with a 95% confidence interval of 1.6–2.6 nM. The host cells (HFF) were also affected by this compound. However, the IC_50_ was 1.3 µM (0.8–2.3 µM confidence interval), thus three orders of magnitude higher (Figure 1).

The ultrastructural alterations in *T. gondii* RH tachyzoites cultured in human foreskin fibroblasts induced by peptide 6027 treatment were visualized by TEM. In non-treated cultures, tachyzoites proliferated intracellularly within parasitophorous vacuoles, each surrounded by a parasitophorous vacuole membrane (Figure 2A,C). TEM revealed the typical features of these parasites, such as the apical complex with a conoid, micronemes, rhoptry organelles and dense granules (Figure 2A,B). Tachyzoites contain a single, multi-tubular mitochondrion, which occupies a substantial part of the cytoplasm. However, only parts of the mitochondrion were visible in any given section plane, displaying a relatively electron-dense matrix and tightly packed cristae (Figure 2C–G). Tachyzoites divide by endodyogeny, which resulted in the formation of daughter zoites (Figure 2F).

In cultures that were exposed to 200 nM peptide 6027, structural alterations in vacuoles containing 1–3 tachyzoites were already visible after 6 h (Figure 3A–D). Two types of parasitophorous vacuoles were detected, sometimes even within the same host cell, as seen in Figure 3A,B. In the majority of instances, the interior of the vacuole was of similar appearance as in the non-treated cultures, but in some cases, the vacuole matrix was filled with a seemingly tightly packed electron-dense substance. In most tachyzoites, the mitochondrion, with its characteristic dark matrix, had largely disappeared or was not discernible anymore, and the cytoplasm contained numerous vacuoles, often accumulating material with low electron density. Closer inspection showed that at least some of these vacuoles were the residues of the mitochondrion, with largely dissolved cristae and matrix residues, but still exhibiting a structurally intact membrane (Figure 3C,D). At 24 h of treatment (Figure 3E), parasites largely retained these alterations but nevertheless appeared to be able to undergo endodyogeny and formed daughter zoites that emerged from tachyzoites. In samples fixed and processed after 48 h (Figure 3F), many parasites were observed that were not enclosed by a parasitophorous vacuole anymore, indicating that peptide 6027 interfered with the integrity of the respective membrane.

### 3.2. Peptide 6027 Has Detrimental Effects When Applied to CD1 Mice Orally Infected with T. gondii Oocysts

The anti-*Toxoplasma* activity of peptide 6027 was assessed in an experimental murine model for *T. gondii* oocyst infection based on TgShSp1 as described in Materials and Methods. Mice were checked daily for potential adverse reactions or clinical signs, and while no such signs were noted in the positive (C+) and negative (C−) control groups, mild clinical signs were observed transiently during the first 10 days p.i. in the group that was infected and received peptide 6027. At 4 weeks p.i., mice were sacrificed, and real time PCR-based quantification showed that the peptide 6027-treated animals had an even higher cerebral (*p* < 0.05) parasite load compared to the placebo-treated control animals (Figure 4A). There was no difference in IgG-titers between the two (Figure 4B).

IFN-γ responses of spleen cells to ConA stimulation were lower in splenocyte cultures of the peptide 6027 group compared to the other groups (*p* = 0.0001), while no or only very little IFN-γ was measured in the supernatants of LPS-stimulated spleen cells (*p* = 0.0207). However, no difference in IFN-γ production was noted when splenocytes were stimulated with *T. gondii* extract (*p* = 0.1949) (Figure 5). Overall, this experiment showed that in vivo treatment did not mirror the excellent in vitro efficacy results and that peptide 6027 treatment was not only ineffective but also exacerbated *T. gondii* infection.

### 3.3. Peptide 6027 Affects the Proliferative Capacity and Viability of Murine B Cells

The effects of peptide 6027 treatment on murine immune cells were investigated in vitro. Splenocytes represent a mixed population of different immune cells, and, to study how this peptide differentially affected T and B cells, splenocytes were treated with ConA to stimulate T cells and LPS to stimulate B cells, respectively, either in the presence or the absence of peptide 6027. As seen in Figure 6, peptide 6027 treatments at a concentration range of 0.1–0.5 µM did affect either proliferation or viability of T cells, while cyclosporin A, a known T cell inhibitor, exerted a pronounced effect (Figure 6A). However, the viability of B cells, as well as B cell proliferation, were impaired in a dose-dependent manner, reaching 50% inhibition for both when applied at 2 µM (Figure 6B). This viability impairment is in a similar range as that observed for HFF (Figure 1).

### 3.4. DAC Proteomes of Toxoplasma gondii Tachyzoites and Mouse Spleens

The pronounced effects of peptide 6027 against *T. gondii* tachyzoites, as well as mouse splenocytes, prompted us to identify peptide 6027 binding proteins via differential affinity chromatography (DAC). Mass spectrometry analysis of the proteomes obtained after DAC of cell-free extracts of *T. gondii* RH tachyzoites resulted in the identification of 17.

16 unique peptides matching to 303 proteins. The complete dataset is given in Appendix A. The corresponding numbers for the proteomes obtained by the DAC of *Mus musculus* spleens were 16,117 unique peptides matching to 1918 proteins. The complete dataset is given in Appendix A.

Unbiased analysis of the dataset by principal component analysis (PCA) demonstrated that the differential affino-proteomes eluted from mock, peptide 6027 and peptide 21,358 columns were located in non-overlapping clusters separated by both principal components. This was true for both organisms and with both Top3 and LFQ algorithms (Figure 7).

A more detailed analysis of the dataset obtained from *T. gondii* tachyzoites revealed 269 binding proteins in peptide 6027 eluates only. Five proteins were found in all eluates and 26 proteins in eluates from both mock and peptide 6027 columns; one protein was found in eluates from peptide 21,358 column only, and 2 were found in both peptide 6027 and peptide 21,358 eluates, as depicted in the Venn diagram presented as Figure 8.

The twenty most abundant binding proteins specific for peptide 6027 column eluates are listed in Table 1. The complete list of the peptide 6027 specific affino-proteome is given as Appendix A.

Thus, in *T. gondii* tachyzoite cell-free extracts, the Sec61beta family protein encoded by ORF TGRH88_008910 was the most abundant peptide 6027 specific binding protein representing 9% of the affino-proteome, followed by GRA12 and a eukaryotic porin protein (Table 1).

In mouse spleen cell-free extracts, 645 peptide 6027 specific binding proteins could be identified. More than 1000 more proteins binding to the peptide 6027 column were identified in mock and peptide 21,358 eluates as well (Figure 7). The twenty most abundant binding proteins are listed in Table 2. Overall these most abundant proteins from spleen cell extracts exhibited similar and lower relative abundance percentages (1–2.1%) compared to the top twenty categorization from *T. gondii,* which exhibited a higher relative abundance for the first five proteins (3.1–9%) as compared to the rest (1.2–2%).

In a next step, analysis of the protein–protein interaction network of the entire affino-proteomes was performed. In the peptide 6027 specific affino-proteome from *T. gondii* tachyzoites, 154 nodes interacting via 852 edges could be identified. This numbers was significantly higher than the number of expected edges without forming clusters. Within this network, two major clusters, one related to ATP synthesis and one related to the 26S-proteasome, could be identified (Figure 9).

Analysis of the peptide 6027-specific affino-proteome obtained from mouse spleen cell-free extracts led to the identification of 525 nodes interacting via 2272 edges. Here again, the number of edges was significantly higher than the number of edges without clusters. Within the resulting network, five major clusters could be identified (Figure 10).

These clusters comprised proteins involved in cell division, RNA processing, protein secretion and processing, as well as proteins from complex I of the respiratory chain. Thus, in both *T. gondii* as well as splenocytes, clusters with functions essential for the maintenance of cellular homeostasis were identified.

### 3.5. More Detailed Analysis of Proteome Data Reveals Putative Drug Targets of Peptide 6027 in T. gondii and Murine Spleen Cells

Among the twenty most abundant specific 6027 binding proteins shown in Table 1, sixteen could be assigned to three distinct clusters based on their subcellular localization; together these three clusters account for 45% of the entire *T. gondii* 6027-affino-proteome. The most abundant cluster represents 27% of the relative abundance and involves twelve mitochondrial proteins. Among those, TgApiCOX13/16/18/19/23/26/30/35 and TgCOX2b are located in the inner mitochondrial membrane, and the eukaryotic porin protein is found in the outer membrane. In addition, proteomics identified the mitochondrial 40 S ribosomal subunit RPS17 and a putative ubiquinol-cytochrome c oxidoreductase. The second largest cluster accounts for 11% relative abundance and is associated with the endoplasmic reticulum (ER), including the Sec61beta family protein and the chaperonin protein BiP, a member of the heat shock protein 70 (Hsp70) family. The Sec61beta family protein is associated with the ER membrane, while BiP is an intrinsic ATPase within the ER. The third group involves two dense granule proteins; GRA 12 and GRA 8, which together account for 8% of the total 6027 affino-proteome. The facilitative glucose transporter GT1, ranked as the fourteenth most abundant protein, is an integral protein of the plasma membrane. A graphic overview is given as Appendix A.

The twenty most abundant peptide 6027-binding proteins proteins listed in Table 2 represent 29% of the total relative abundance. Among these twenty hits, nine proteins (Q9D958, O08547, Q8R5J9, Q9D8V7, Q9DCF9-2, P61166, Q9JKW0, Q9R0P6 and Q9R0Q3), mostly proteolytic subunits of the signal peptidase complex, were assigned as endoplasmic reticulum membrane proteins, representing 14% of the spleen 6027-affino-proteome. The three mitochondrial proteins that were identified account for 5% relative abundance. BRI3-binding protein and cytochrome c oxidase subunit 7C are located in the outer and inner mitochondrial membrane, respectively, whereas the coiled-coil-helix-coiled-coil-helix domain-containing protein 5 is located at the intermembrane space. Plasma membrane proteins represent 10% relative abundance. Among those, the two integral membrane proteins, namely lymphocyte antigen 6D and B-lymphocyte antigen CD20, are specific B-lymphocyte surface proteins. A graphic overview is given as Appendix A.

## 4. Discussion

Previous studies had demonstrated that the leucinostatin-derived antimicrobial peptide 6027 was highly effective in vitro against protozoan parasites including *Trypanosma brucei*, *Leishmania donovani* and *Plasmodium falciparum*. Based on structure–activity analysis with a series of leucinostatin derivatives with respect to efficacy, ultrastructural alterations in *T. brucei*, artificial membrane models and membrane potential, it was concluded that the main mechanism of action of this synthetic leucinostatin was in the destabilization of the inner mitochondrial membrane.

Herein, we show that peptide 6027 is also highly effective against the apicomplexan parasite *T. gondii*, by inhibiting tachyzoite proliferation in vitro with an IC_50_ in a similar range as that reported for *T. b. rhodesiense* bloodstream forms [17], most likely by hitting the mitochondrion as the main subcellular target affecting its structural integrity within a few hours of treatment, similar to what was previously observed with *T. brucei* bloodstream forms [17]. Thus, the destabilization of the inner mitochondrial membrane could be a common mechanism of action. Another membrane that was highly affected in many *T. gondii*-infected cells, albeit after more prolonged treatment of 48 h, was the parasitophorous vacuole membrane. This resulted in parasites that were located more or less freely in the cytoplasm. Similarly damaged parasitophorous vacuole membranes had been observed earlier in HFF infected with the closely related parasite *Neospora caninum* and treated with nitazoxanide and other thiazolides [25].

Based on these promising in vitro results, we embarked on a small-scale animal study in the *Toxoplasma* oocyst infection model. The results of this in vivo study, however, showed that subcutaneous application of peptide 6027 in *T. gondii*-infected mice exerted detrimental effects, in that treated mice exhibited a higher cerebral parasite load compared to their non-treated infected counterparts. In addition, peptide 6027-treated mice exhibited mild transient clinical signs during the first 10 days post-infection. This indicated that peptide 6027 promoted, rather than inhibited, the infection process in vivo, which in turn could be due to a potential interference in the immune response. As IFN-γ is the major cytokine known to confer protective immunity against *T. gondii* infection [26], we investigated the production of this cytokine in splenocyte cultures obtained from these experimentally infected mice. While ConA-stimulated splenocytes obtained from peptide 6027-treated mice, no matter whether they were infected or not, displayed a reduced IFN-γ production compared to the non-treated controls, no difference was noted in the two splenocyte populations from infected mice when they were stimulated with *T. gondii* extract. Clearly, a much wider range of cytokines and immune cells should be investigated to elucidate the characteristics of a potentially differential immune response in treated versus non-treated mice.

However, the exposure of naîve splenocytes stimulated with ConA and LPS treated with peptide 6027 revealed that B cells, but not T cells, were affected in their proliferative responses and viability in a dose-dependent manner. This means that the treatment itself could have potentially impacted on the immune response to infection. Thus, the action of this peptide is very likely not specific for protozoans but probably also affects mouse splenocytes. A common, unspecific mode of action may reside in the uncoupling of membrane potentials, due to ionophore activities, as observed for the mother compound leucinostatin in earlier studies [27,28,29].

Our results obtained with differential affinity chromatography suggest, however, that membrane potential uncoupling may not be the only mode of action of leucinostatin peptides and analogues. In fact, we have identified proteins of the respiratory chain such as cytochrome c oxidase (complex IV) subunits [30] among the most abundant peptide 6027 specific binding proteins from both organisms and a putative coenzyme Q-cytochrome c oxidoreductase (complex III [31]) among the most abundant binding proteins from *T. gondii*. In fact, many of the *T. gondii* proteins annoted as unspecified products in Table 1 are actually subunits of the *T. gondii* cytochrome oxidase (TgApiCox25) complex [32]. Precisely these eight subunits, together with another four, namely TgApiCox24/25 and TgCox2a/5b (also identified in our proteomics analysis and annoted as TGRH88_067080, TGRH88_025670, TGRH88_046190 and TGRH88_021840, respectively, see Appendix A), form a ~600 kDa protein complex in the inner mitochondrial membrane (IMM). Among the TgApiCox25 complex subunits, nine (TgApiCox16/18/19/23/24/25/26/30 and 35) are restricted to the apicomplexan lineage, with TgApiCox16 being specific to *T. gondii* [32].

Eukaryotic porin was identified as the third most abundant 6027-binding protein in *T. gondii.* In contrast to the ApiCox25 complex, porin protein, also known as voltage-dependent anion channel (VDAC), is an outer mitochondrial membrane protein (OMM). In *Toxoplasma*, the depletion of VDAC leads to abnormalities in mitochondrial morphology and physiology and the appearance of “ball shaped” mitochondria with impaired metabolism and protein import [33]. Thus, the fact that the most significant protein binding to peptide 6027 happens via mitochondrial proteins may explain why the major alterations detected by TEM are found in the mitochondria.

Likewise, in mouse spleen extracts, three mitochondrial proteins were within the top 20 proteins binding to peptide 6027, of which only cytochrome c oxidase subunit 7C is located in the IMM. A previous study on leucinostatins A and B had demonstrated the inhibition of oxidative phosphorylation in rat liver mitochondria [34]. In the present study, we could not identify components of the electron transport chain nor ATP synthase complex subunits among the splenocyte peptide 6027 binding proteins, but STRING interaction analysis suggested proteins clustering within the NADH-dehydrogenase complex I [35] in the 6027-specific affino-proteome from mouse spleen. These results indicate that mitochondrial damage followed by the disruption of the energy homeostasis in treated cells may be the result of the inhibition of respiratory chain components and/or ATP synthesis.

Besides mitochondrial proteins, two *T. gondii*-dense granule proteins, namely GRA12 and GRA8, were interacting with peptide 6027. GRA12, the second most frequently found protein, is involved in the formation of the intravacuolar network (IVN) within the parasitophorous vacuole (PV) [36] and is considered a major virulence factor required for parasite resistance to interferon gamma [37]. Another GRA protein identified in this study, GRA8, is involved in the formation of the PV [38] and contributes to the organization of the tachyzoite subpellicular cytoskeleton and motility [39]. The binding of peptide 6027 to GRA8 and GRA12 could interfere with their function causing the structural abnormalities of the PV in treated cultures as visualized by TEM.

In both organisms, besides the impairment of mitochondrial function, peptide 6027 may affect other essential cellular compartments and functions by interacting with the protein processing and targeting machinery, as suggested by the protein–protein network analyses of peptide 6027 affino-proteomes. In *T. gondii* extracts, the major peptide 6027-specific binding protein, Sec61 beta, encoded by ORF TgRH88_008910, is part of the Sec61 translocon complex situated in the endoplasmatic reticulum (ER) membrane [40], responsible for the translocation of polypeptides from the ER surface into the lumen. It is therefore not surprising that—according to the CRISPR-CAS knockdown data provided by ToxoDB (www.toxodb.org)—the ortholog of TgRH88_008910 in *T. gondii* ME49, TgME49_211040, is highly fitness-conferring. Members of other complexes involved in protein sorting, namely Sec11, responsible for the cleavage of signal peptides [41], and Sec22, responsible for the ER-to-Golgi transition [42], are found among the most abundant peptide 6027 binding proteins from mouse spleens. Moreover, the 26S proteasome belongs to the clusters of interacting proteins identified within the *T. gondii* affino-proteome. This is in good agreement with earlier studies showing that lactacystin and other proteasome inhibitors block the proliferation of *T. gondii* tachyzoites in vitro [43].

Based on our data, peptide 6027 tends to interact with protein-modifying systems. This may be due to structural similarities between the peptide and targeting sequences in the substrate proteins of these systems. In spleen cells, ER proteins and plasma membrane proteins comprise 14 and 10% of the relatively abundant proteins. Many ER proteins are subunits of the integral ER membrane signal peptidase complexes, which cleave off signal peptides from precursor proteins. In terms of plasma membrane components, two specific B-lymphocytes receptors were among the 20 most abundant peptide 6027 binding proteins. CD20 is a cell-surface tetraspan receptor expressed exclusively on B-lymphocytes playing an important role in the activation and proliferation of human B-cells [44]. By using monoclonal anti-CD20 antibodies, B cell populations including memory B cells in peripheral blood can be decreased [45]. Similarly, the interaction of CD20 with peptide 6027 could affect both humoral and cellular immune responses, thereby causing treatment failure in vivo. In fact, during the course of the Th1 inflammatory response to *T. gondii* infection in mice, B cells partially mediate the increase of IFN-γ secretion by the effector T cells. The B-lymphocytes induce this upregulation via TNF-α expressed on their surface [46]. Our proteomic data are thus in frame with the observation that B-cell viability and proliferation are affected by 6027 in vitro and with the low IFN-γ levels found in 6027-treated mice in vivo.

Since peptides can be easily modified, evolution circles consisting of modeling–in vitro testing—DAC—re-modeling of the peptide could lead to a peptide with increased specificity for the pathogen and decreased adverse side effects for the host, without the use of extensive animal experimentation.

## 5. Conclusions and Outlook

Taken together, our results show that differential affinity chromatography (DAC) coupled to LC/MS constitutes a useful tool to elucidate whether a compound of interest, herein an antimicrobial peptide, interacts with essential pathways in both pathogens and hosts. Since interaction with essential pathways in the host may cause severe side effects, unnecessary animal studies may be avoided. It appears that the identification of binding proteins involved in essential host pathways, such as cell division, gene expression and energy metabolism, should be good criteria for the elimination of the compound from in vivo studies. This approach is versatile since any compound that can be coupled to a matrix can be tested. Suitable controls should include ineffective analogs coupled to the same matrix. The set-up presented here requires extracts with proteins in milligram-amounts. Due to the increasing sensitivity and throughput of MS, the method can be downscaled with respect to the quantity of protein needed and upscaled with respect to the numbers of columns processed in parallel. By using host organ explants or even biopsy material instead of cell cultures, DAC can be performed on the proteomes with which the drug will interact in vivo. As a consequence, by applying this approach in a routine drug development program, it would be possible to avoid animal testing, since adverse side effects can be largely predicted, and thus, to contribute to the 3R principles by replacing and reducing animal studies. For clinical research, the use of biopsy material could be a step forward to personalized medicine.

## Figures and Tables

**Figure 1 biomedicines-10-02675-f001:**
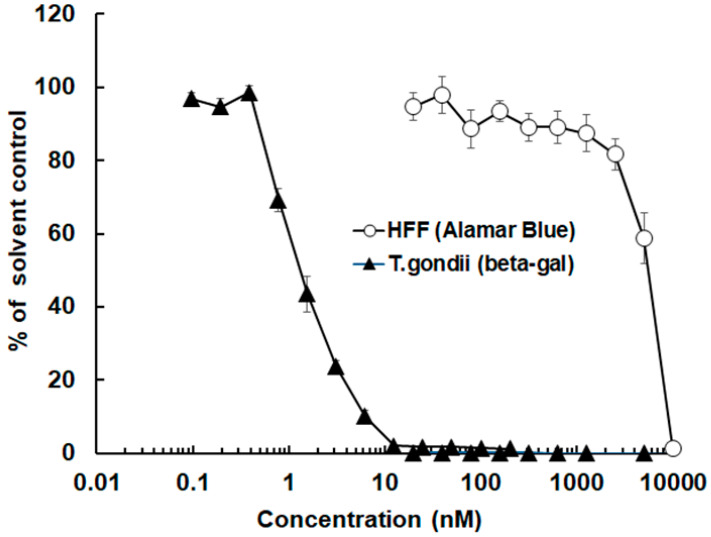
The antimicrobial peptide 6027 inhibits *Toxoplasma gondii* at nanomolar concentrations. The inhibition of *T. gondii* RH tachyzoite proliferation was determined using a beta-galactosidase reporter strain in the presence of a concentration series of peptide 6027 added prior to the infection of confluent HFF by tachyzoites. The impact of peptide 6027 on the vitality of uninfected HFF was assessed via resazurin reduction. Mean values ± SE are indicated for quadruplicates. The x-axis scale is logarithmic at basis 10.

**Figure 2 biomedicines-10-02675-f002:**
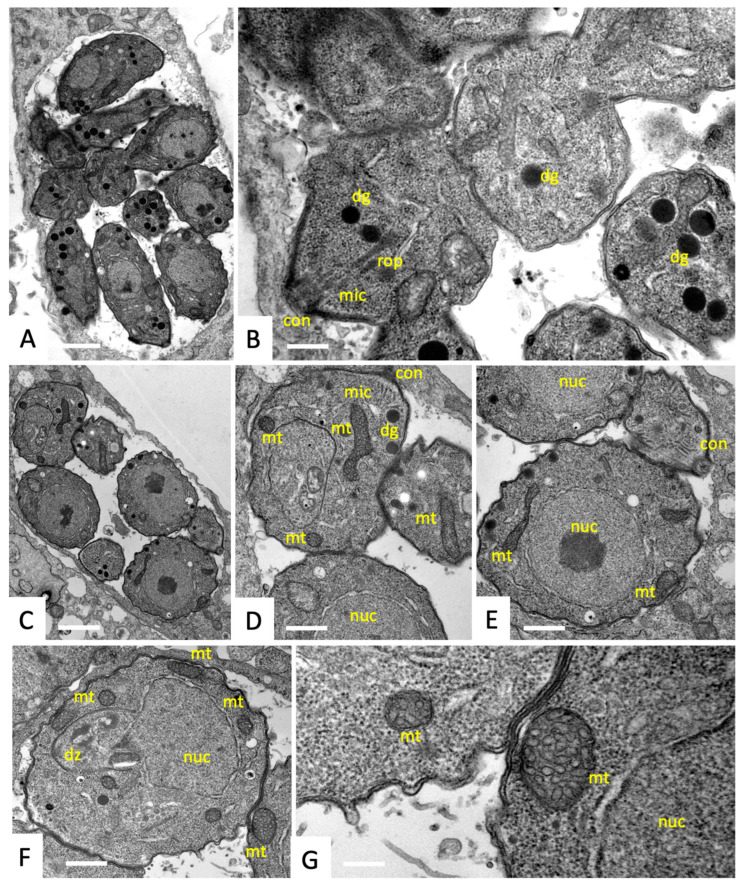
Ultrastructure of *T. gondii* tachyzoites cultured in human foreskin fibroblasts (HFF). Parasites proliferated intracellularly within a parasitophorous vacuole (**A**,**C**), and corresponding higher magnification views are depicted in (**B**) and (**D**–**E**). A tachyzoite undergoing endodyogeny forming a daughter zoite (dz) is shown in (**F**), and structural details of the mitochondrion are shown at higher magnification in (**G**). Notice distinct apicomplexan organelles at the anterior end of tachyzoites such as the conoid (con), rhoptries (rop) and micronemes (mic). Dense granules (dg) and parts of the mitochondrion (mt) are dispersed within the cytoplasm; nuc = nucleus. Bars in (**A**) = 1 µm; (**B**) = 0.35 µm; (**C**) = 1 µm; (**D**) = 0.55 µm; (**E**) = 0.42 µm; (**F**) = 0.55 µm; (**G**) = 0.18 µm.

**Figure 3 biomedicines-10-02675-f003:**
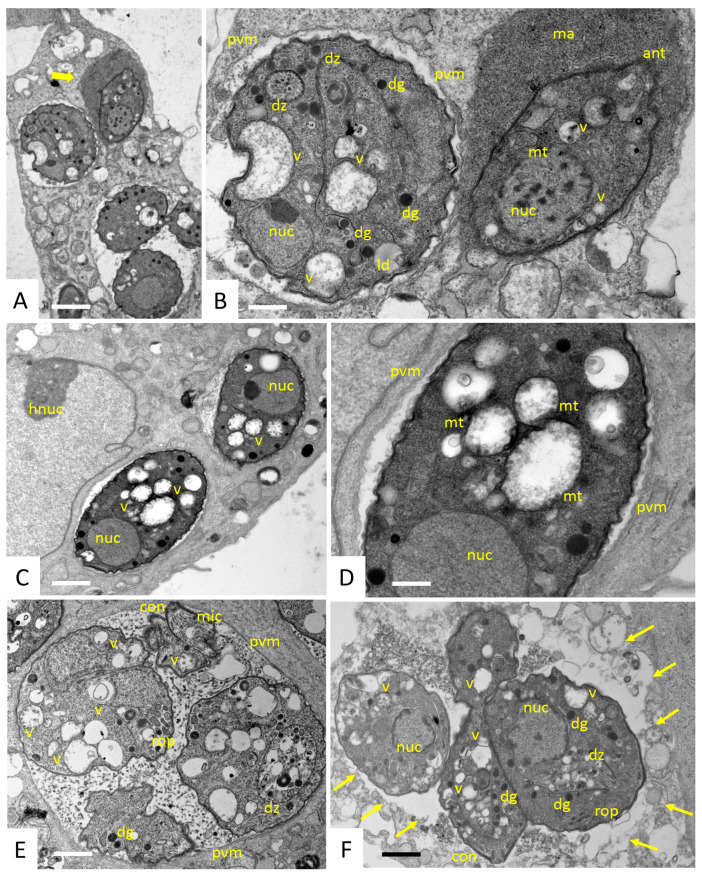
Ultrastructure of *T. gondii* tachyzoites grown in HFF exposed to peptide 6027 during 6 h (**A**–**D**), 24 h (**E**) and 48 h (**F**). (**A**,**C**) are lower-magnification views, while (**B**,**D**) show more detailed views at higher magnification. The bold arrow in A points to a tachyzoite located in a vacuole that is filled with an electron-dense matrix (ma), and the thin arrows in (**F**) indicate the lack of a defined parasitophorous vacuole membrane (pvm) after 48 h of treatment (**F**); ant = anterior end; dz = emerging daughter zoites; dg = dense granules; hnuc = host nucleus; rop = rhoptries; mic = micronemes; nuc = nucleus; con = conoid; v = vacuolization; mt = mitochondrial residues; ld = lipid droplets. Bars in (**A**) = 1.1 µm; (**B**) = 0.35 µm; (**C**)= 0.75 µm; (**D**) = 0.28 µm; (**E**,**F**) = 0.75 µm.

**Figure 4 biomedicines-10-02675-f004:**
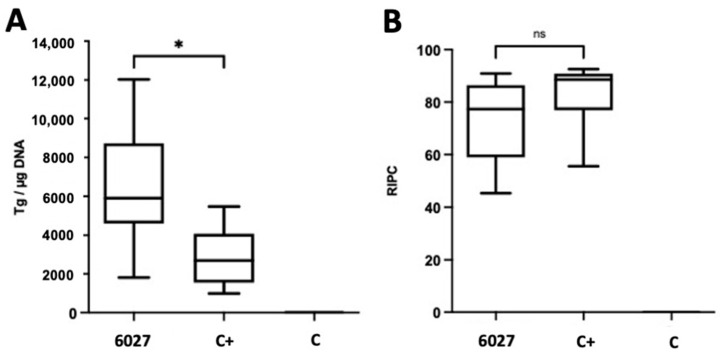
Cerebral parasite load (**A**) and IgG-titers (**B**) in CD1 mice infected with *T. gondii* oocysts and either treated with peptide 6027 (3 mg/kg/day for 5 days) or treated with placebo only (C+). C− was not infected and only placebo treated; * indicates *p* < 0.05; ns = not significant.

**Figure 5 biomedicines-10-02675-f005:**
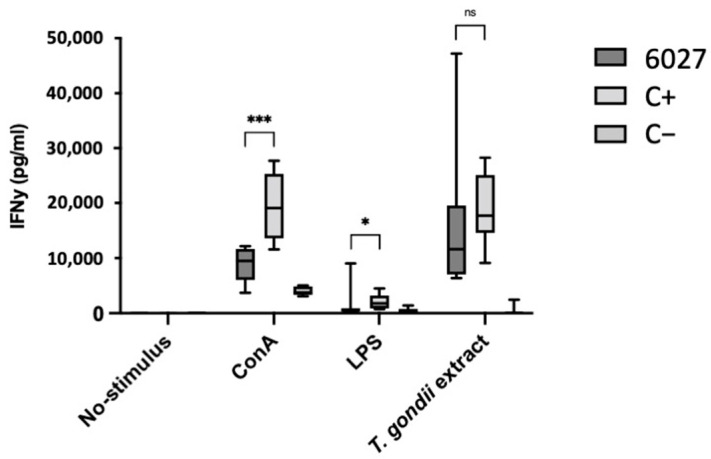
IFN-γ responses of splenocyte cultures in CD1 mice infected with *T. gondii* oocysts and either treated with peptide 6027 (3 mg/kg/day for 5 days) or treated with placebo only (C+). C− was not infected and only placebo treated; ns indicates no statistically significant difference; * = *p* < 0.05; *** indicates *p* = 0.0001; ns, not significant.

**Figure 6 biomedicines-10-02675-f006:**
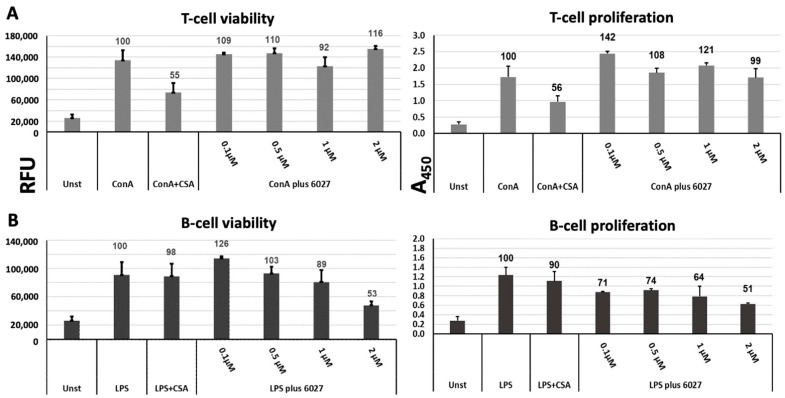
Effects of peptide 6027 on murine T cells (**A**) and B cells (**B**) with respect to viability and proliferation in vitro; 96-well plates were seeded with splenocytes obtained from murine spleen (2 × 10^6^ cells/mL, 100 µL/well), and were exposed to ConA (5 µg/mL) or LPS (10 µg/mL). Peptide 6027 was added at 0.1, 0.5, 1 and 2 µM, respectively, and cultivation was carried out for 48 h at 37 °C/5% CO_2_. Viability was assessed by resazurin reduction and is given as relative fluorescence units (RFU); the proliferation of cells was measured by BrdU ELISA and is given as absorption at 450 nm (A_450_).

**Figure 7 biomedicines-10-02675-f007:**
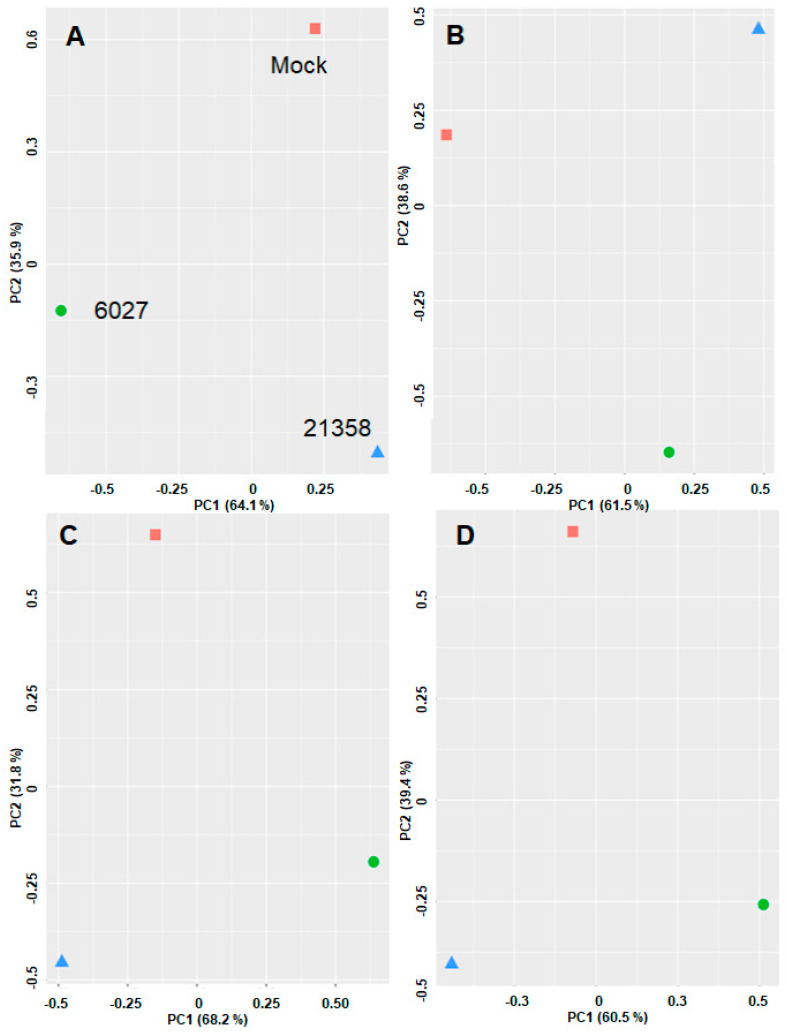
Principal component analysis of affino-proteome data sets from *T. gondii* RH tachyzoites (**A**,**B**) and from *Mus musculus* spleens (**C**,**D**). Cell-free extracts were prepared and subjected to differential affinity chromatography on mock (red square), peptide 21,358 (blue triangle) or peptide 6027 (green circle) columns followed by mass spectrometry as described in the Materials and Methods. (**A**,**C**), Top3 data; (**B**,**D**), LFQ data. X-axis, principal component 1; Y-axis, principal component 2.

**Figure 8 biomedicines-10-02675-f008:**
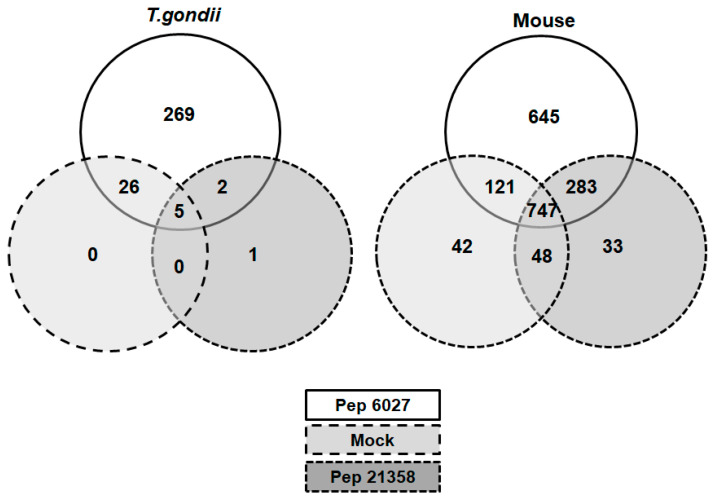
Venn diagram detailing the number of proteins identified in eluates from peptide 6027 (white), peptide 21,358 (light grey) or mock (dark grey) columns loaded with cell-free extracts from *T. gondii* RH tachyzoites or mouse spleens. The complete datasets are given in Appendix A, respectively.

**Figure 9 biomedicines-10-02675-f009:**
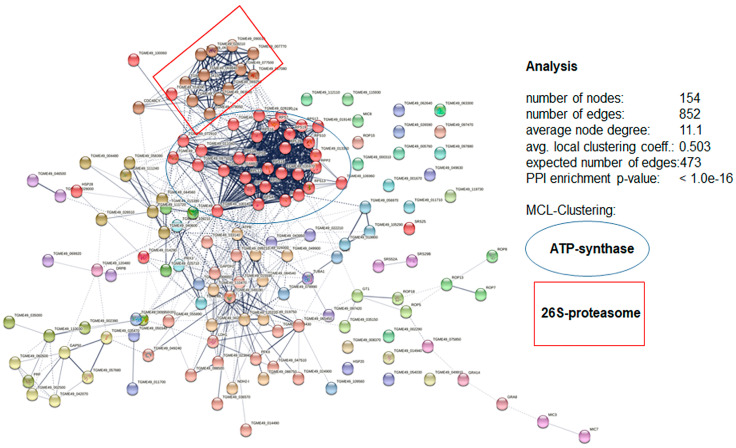
Protein–protein interaction network of proteins specifically binding to peptide 6027 in cell-free extracts of *T. gondii* RH tachyzoites based on the list of proteins given in Appendix A. Two major clusters, namely ATP-synthase (blue ellipse) and the 26S-proteasome (red rectangle). are highlighted. The interaction network was created by the STRING knowledgebase and software tool from the Swiss Institute of Bioinformatics (www.expasy.org).

**Figure 10 biomedicines-10-02675-f010:**
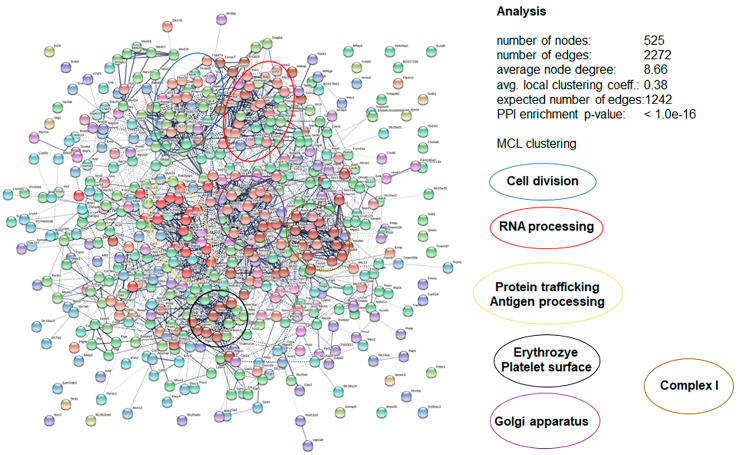
Protein–protein interaction network of proteins specifically binding to peptide 6027 in cell-free extracts of mouse spleens. Five clusters, namely cell division (blue), RNA processing (red), protein trafficking and antigen processing (yellow), erythrocyte and platelet surface (black), golgi apparatus (pink) and respiratory chain complex I (brown) are surrounded by ellipses. The interaction network was created by the STRING knowledge base and software tool from the Swiss Institute of Bioinformatics (www.expasy.org).

**Table 1 biomedicines-10-02675-t001:** List of the twenty most abundant peptide 6027-specific binding proteins in *T. gondii* RH tachyzoite cell-free extracts. The ORF numbers correspond to Toxo DB entries (www.toxodb.org, accessed 17 January 2022). Their relative abundance is given as a parameter taken from the dataset (see Appendix A), as well as the percentage of the peptide 6027 specific binding proteins. The full list of proteins is given in Appendix A. The potential identity of some unspecified products was determined by BLAST searches and is provided in brackets.

ORF	Annotation	Relative Abundance (×10^3^) (%)	
TGRH88_008910	Sec61beta family protein	33.2	9.0
TGRH88_013490	Dense granule protein GRA12	22.8	6.2
TGRH88_067710	Eukaryotic porin protein	20.0	5.4
TGRH88_002590	Unspecified product (TgApiCox30)	14.5	3.9
TGRH88_000580	Unspecified product (TgApiCox18)	11.7	3.2
TGRH88_067940	14-3-3 protein	11.3	3.1
TGRH88_061860	Unspecified product (TgApiCox19)	7.5	2.0
TGRH88_004320	Zinc finger CDGSH-type domain-containing protein	6.9	1.9
TGRH88_006890	Putative ubiquinol cytochrome c oxidoreductase	6.8	1.8
TGRH88_018120	Unspecified product (TgApiCox26)	6.6	1.8
TGRH88_074430	Unspecified product (TgApiCox35)	6.5	1.8
TGRH88_067370	Unspecified product	6.5	1.8
TGRH88_081780	Putative calmodulin	6.2	1.7
TGRH88_040180	Facilitative glucose transporter GT1	6.0	1.6
TGRH88_051090	Chaperonin protein BiP	5.9	1.6
TGRH88_003600	Dense granule protein GRA8	5.8	1.6
TGRH88_050010	Putative cytochrome C oxidase subunit IIb (TgCox2b)	5.2	1.4
TGRH88_022800	Ribosomal protein RPS17	4.9	1.3
TGRH88_012110	Unspecified product (TgApiCox16)	4.6	1.3
TGRH88_068380	Cg8 family protein (TgApiCox23)	4.6	1.2

**Table 2 biomedicines-10-02675-t002:** List of most abundant 6027-specific binding proteins in mouse spleen cell-free extracts. The IDs correspond to Uniprot entries (www.uniprot.org). The ORF numbers correspond to Uniprot entries (www.uniprot.org). Their relative abundance is given as a parameter taken from the dataset (see Appendix A), as well as a percentage of the peptide 6027 specific binding proteins. The full list of proteins is given in Appendix A.

IDs	Annotation	Relative Abundance (rAbu) (%)	
Q9CQP3	Coiled-coil-helix-coiled-coil-helix domain-containing protein 5	1290	2.1
Q9D958	Signal peptidase complex subunit 1	1231	2.0
O08547	Vesicle-trafficking protein SEC22b	1186	1.9
Q8R5J9	PRA1 family protein 3	1162	1.9
P17665	Cytochrome c oxidase subunit 7C, mitochondrial	1143	1.8
P35459	Lymphocyte antigen 6D	1141	1.8
Q08857	Platelet glycoprotein 4	1078	1.7
Q9D8V7	Signal peptidase complex catalytic subunit SEC11C	1023	1.6
P20491	High affinity immunoglobulin epsilon receptor subunit gamma	1004	1.6
Q8BXV2	BRI3-binding protein	945	1.5
P19437	B-lymphocyte antigen CD20	935	1.5
P97370	Sodium/potassium-transporting ATPase subunit beta-3	907	1.4
Q9DCF9-2	Isoform 2 of Translocon-associated protein subunit gamma	835	1.3
P61166	Transmembrane protein 258	821	1.3
Q9JKW0	ADP-ribosylation factor-like protein 6-interacting protein	821	1.3
Q9R0P6	Signal peptidase complex catalytic subunit SEC11A	819	1.3
P54116	Stomatin	680	1.1
Q9D3P8	Plasminogen receptor (KT)	601	1.0
Q9R0Q3	Transmembrane emp24 domain-containing protein 2	600	1.0
P46638	Ras-related protein Rab-11B	573	0.9

## Data Availability

Data are made available as Appendix A datasets (see above).

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
