# Peer review of "Differential Affinity Chromatography Coupled to Mass Spectrometry: A Suitable Tool to Identify Common Binding Proteins of a Broad-Range Antimicrobial Peptide Derived from Leucinostatin"

_biomedicines, 2022, doi:10.3390/biomedicines10112675_

Round 1

Reviewer 1 Report

The authors presented the usefulness of DAC coupled to LC/MS with peptide 6027 which showed an efficacy in vitro but adverse effect in vitro in mice.

With this version of manuscript, The reviewer think it is very hard to accept for the publication in Biomedicines for the reason below.

The authors mentioned their recent studies, ref 14 and 15, where they successfully revealed the cellular and molecular target of the potential inhibitors. The format and flow of this manuscript are very similar to those of the previous publications. The main difference is the molecule, peptide 6027, is not efficacious in vivo in this manuscript. Thus, the reviewer think the study has little meaningful and novel findings.

Author Response

Reviewer 1 (Our comments and corrections in the manuscript are in blue)

The authors presented the usefulness of DAC coupled to LC/MS with peptide 6027 which showed an efficacy in vitro but adverse effect in vitro in mice.

With this version of manuscript, The reviewer think it is very hard to accept for the publication in Biomedicines for the reason below.

The authors mentioned their recent studies, ref 14 and 15, where they successfully revealed the cellular and molecular target of the potential inhibitors. The format and flow of this manuscript are very similar to those of the previous publications. The main difference is the molecule, peptide 6027, is not efficacious in vivo in this manuscript. Thus, the reviewer think the study has little meaningful and novel findings.

Our response:

We do not agree with this assessment. The goal of this manuscript is not to advertise a novel compound with excellent activities against T. gondii, but to highlight a way to avoid animal experimentation by carrying out DAC combined with MS/proteomics. By using DAC, we may identify essential pathways in both pathogens and hosts thereby minimizing the use of animal studies and perhaps even failures at later stages of drug development. We have added this statement to the Conclusions (lane 594 f.)

Reviewer 2 Report

Comment:

1.     TEM images shown with no scale on the bars.

2.     Typo should be corrected in the SI.

3.     DAC data with Mass should be shown in the SI.

Author Response

Reviewer 2 (Our corrections are in green)

  1. TEM images shown with no scale on the bars.
  2. Typo should be corrected in the SI.
  3. DAC data with Mass should be shown in the SI.

These corrections have been made in Figs. The SI tables do actually contain all relevant data (https://zenodo.org/record/7053105#.Yxb9HC223ow)

.

Reviewer 3 Report

The manuscript by Muller J. et al. investigated the antimicrobial activity of a leucinostatin derivative peptide (peptide 6027) against Toxoplasma gondii in vitro and in vivo. While peptide 6027 showed high antimicrobial activity against T. gondii in vitro, it failed to show efficacy in vivo as it produced higher cerebral parasite load compared to control and impaired the proliferation and viability of B cells. The authors then used differential affinity chromatography (DAC) coupled to mass spectrometry (MS) to identify the proteins in T. gondii and host cell that specifically bind to peptide 6027. They found that peptide 6027 bind mostly to mitochondrial and endoplasmic reticulum proteins in both the parasite and host cells, which are involved in vital cellular processes such as respiration and protein processing and targeting. The authors posited that this could be the reason why peptide 6027 was ineffective in vivo. Overall, the results were presented systematically and the proteome data was thoroughly analyzed and explained. The approach presented by the authors would be beneficial in the design of therapeutics as it will minimize the unnecessary use of animal studies.

I only have a few minor comments that will help improve the manuscript:

1. The authors were recommending DAC coupled to mass spectrometry as a technique to use in drug development to test whether a potential drug has possible adverse side effects in vivo before even doing the animal studies. What do the authors think would be a good or acceptable criterion after performing DAC and proteome analysis to be able to say that a potential drug is worth proceeding to animal studies? Can the authors provide some caveats or limitations of the technique in the Conclusions and outlook section?

2. In Figures 4 and 5, the caption includes the legend/annotation for the p value (like ns, *, **, ***) but they were not seen in the actual figure.

3. Figure 6 is missing the y-axis labels.

4. Some typographical errors:

A. Page 2, line 74: a class of broad

B. Page 7, line 261: higher magnification views are depicted in B and D-E, respectively.

C. Page 10, Figure 5: IFNy (instead of INFy)

D. Page 10, line 327: stimulate T cells and and LPS… either in the presence or absence

E. Page 14, line 392: eluates, as well (Figure 8)

F. Page 16, line 435: drug targets of peptide 6027

G. Page 18, line 526: Eukaryotic porin

H. Page 19, line 577: important role in the regulation…

Author Response

The manuscript by Muller J. et al. investigated the antimicrobial activity of a leucinostatin derivative peptide (peptide 6027) against Toxoplasma gondii in vitro and in vivo. While peptide 6027 showed high antimicrobial activity against T. gondii in vitro, it failed to show efficacy in vivo as it produced higher cerebral parasite load compared to control and impaired the proliferation and viability of B cells. The authors then used differential affinity chromatography (DAC) coupled to mass spectrometry (MS) to identify the proteins in T. gondii and host cell that specifically bind to peptide 6027. They found that peptide 6027 bind mostly to mitochondrial and endoplasmic reticulum proteins in both the parasite and host cells, which are involved in vital cellular processes such as respiration and protein processing and targeting. The authors posited that this could be the reason why peptide 6027 was ineffective in vivo. Overall, the results were presented systematically and the proteome data was thoroughly analyzed and explained. The approach presented by the authors would be beneficial in the design of therapeutics as it will minimize the unnecessary use of animal studies.

I only have a few minor comments that will help improve the manuscript:

The authors were recommending DAC coupled to mass spectrometry as a technique to use in drug development to test whether a potential drug has possible adverse side effects in vivo before even doing the animal studies. What do the authors think would be a good or acceptable criterion after performing DAC and proteome analysis to be able to say that a potential drug is worth proceeding to animal studies?

We think that the identification of binding proteins involved in essential host pathways such as cell division, gene expression, energy metabolism should be good criteria for the elimination of the compound from in vivo studies. We have added this information in the conclusions (lane 595 ff).

 Can the authors provide some caveats or limitations of the technique in the Conclusions and outlook section?

As said in the text, DAC is highly versatile, if the compounds of interest can be coupled to a suitable matrix. Moreover, suitable controls should include ineffective analogs coupled to the same matrix. The set-up, which we have used, requires extracts with proteins in mg-amounts. A downscaling should, however, be possible.

These statements have been included to the Conclusions (lane 595 ff.)

  1. In Figures 4 and 5, the caption includes the legend/annotation for the p value (like ns, *, **, ***) but they were not seen in the actual figure.

Thank you for this comment, this is a mistake, and the information was added into the figures and the figure legends were adjusted.

  1. Figure 6 is missing the y-axis labels.

This has been introduced

Some typographical errors:

  1. Page 2, line 74: a class of broad
  2. Page 7, line 261: higher magnification views are depicted in B and D-E, respectively.
  3. Page 10, Figure 5: IFNy (instead of INFy)
  4. Page 10, line 327: stimulate T cells and and LPS… either in the presence or absence
  5. Page 14, line 392: eluates, as well (Figure 8)
  6. Page 16, line 435: drug targets of peptide 6027
  7. Page 18, line 526: Eukaryotic porin
  8. Page 19, line 577: important role in the regulation
  9. Page 19, line 577: important role in the regulation

We have corrected the MS accordingly.

Round 2

Reviewer 1 Report

I think the revised version of the manuscript is acceptable for the publication in Biomedicines.